# Phosphatidic Acid in Plant Hormonal Signaling: From Target Proteins to Membrane Conformations

**DOI:** 10.3390/ijms23063227

**Published:** 2022-03-17

**Authors:** Yaroslav Kolesnikov, Serhii Kretynin, Yaroslava Bukhonska, Igor Pokotylo, Eric Ruelland, Jan Martinec, Volodymyr Kravets

**Affiliations:** 1V.P. Kukhar Institute of Bioorganic Chemistry and Petrochemistry National Academy of Sciences of Ukraine, Murmanska Street, 1, 02660 Kiev, Ukraine; kolesnikov@bpci.kiev.ua (Y.K.); sergey.bpci@gmail.com (S.K.); yasya.yaroslavka@gmail.com (Y.B.); pokotylo@bpci.kiev.ua (I.P.); 2Sorbonne Universités, Génie Enzymatique et Cellulaire, UMR CNRS 7025, Université de Technologie de Compiègne, CEDEX, 60203 Compiègne, France; eric.ruelland@u-pec.fr; 3Institute of Experimental Botany of the Czech Academy of Sciences, Rozvojová 263, 165 00 Prague, Czech Republic

**Keywords:** phospholipase, diacylglycerol kinase, phospholipid, phosphatidic acid, signal transduction, targets, biologically active substance, autophagy

## Abstract

Cells sense a variety of extracellular signals balancing their metabolism and physiology according to changing growth conditions. Plasma membranes are the outermost informational barriers that render cells sensitive to regulatory inputs. Membranes are composed of different types of lipids that play not only structural but also informational roles. Hormones and other regulators are sensed by specific receptors leading to the activation of lipid metabolizing enzymes. These enzymes generate lipid second messengers. Among them, phosphatidic acid (PA) is a well-known intracellular messenger that regulates various cellular processes. This lipid affects the functional properties of cell membranes and binds to specific target proteins leading to either genomic (affecting transcriptome) or non-genomic responses. The subsequent biochemical, cellular and physiological reactions regulate plant growth, development and stress tolerance. In the present review, we focus on primary (genome-independent) signaling events triggered by rapid PA accumulation in plant cells and describe the functional role of PA in mediating response to hormones and hormone-like regulators. The contributions of individual lipid signaling enzymes to the formation of PA by specific stimuli are also discussed. We provide an overview of the current state of knowledge and future perspectives needed to decipher the mode of action of PA in the regulation of cell functions.

## 1. Introduction

Phosphatidic acid (PA) is a minor membrane glycerolipid composed of glycerol-3-phosphate and two fatty acid chains. PA is an intermediate and regulator of glycerolipid metabolism in yeast, animals and plants [1,2,3,4]. Under basal conditions, PA is formed de novo in the endoplasmic reticulum and plastids mainly by acylation of glycerol-3-phosphate catalyzed by specific acyltransferases [5]. PA is also an important lipid second messenger in cells, acting in part through interaction with various effector proteins [6,7]. The level of PA, like that of most phospholipids in cells, is highly dynamic and changes specifically, rapidly (seconds to minutes) and transiently (i.e., eventually reverting to basal level) in response to various extracellular, hormonal or stress stimuli [8,9]. These changes are made possible by various phospholipases, lipid kinases and phosphatases. The generation of signaling PA occurs via the activation of phospholipase D (PLD), which cleaves structural membrane lipids (e.g., phosphatidylcholine) to generate PA and free headgroups (e.g., choline) [10,11,12,13] (Figure 1). Signaling PA can also be produced by the phosphorylation of diacylglycerol (DAG, Figure 1) in a reaction catalyzed by diacylglycerol kinase (DGK) [14,15,16]. The formation of DAG itself is the result of the activity of two classes of phospholipases C: one that hydrolyzes phosphatidylinositol phosphates (phosphatidylinositol-specific phospholipase C, PI-PLC) or one that hydrolyzes structural lipids (phosphatidylcholine-specific or non-specific phospholipase C – PC-PLC or NPC) [17,18]. As a part of lipid turnover, PA can be dephosphorylated back into DAG by lipid phosphate phosphatases (LPP) [19,20]. 

Each class of enzymes involved in the production of PA is represented by multiple isoforms (isoenzymes) [4,21]. Regulation of their activity by extracellular agonists may represent a specific mechanism to control lipid signaling in cells [9,22]. Changes to activity or abundance of particular isoenzymes acting in PA production may be essential for stress tolerance [23,24] or growth/developmental responses [25,26,27,28].

In cell membranes, PA molecules are not compositionally uniform. They are represented by different molecule classes (molecular species) depending on the length and degree of desaturation of fatty acid moieties they incorporate [29]. Accordingly, PA produced in the course of cell signaling is also heterogeneous. *At*PLDα1 activity stimulated by phytohormone abscisic acid (ABA) leads to 34:2-PA, 34:3-PA, 36:3-PA, 36:4-PA, 36:5-PA and 36:6-PA production [30]. The profile of PA molecules generated by *At*PLDδ is different: this enzyme additionally generates 34:1-PA and 36:2-PA, but not 36:3-PA and 36:6-PA [31]. Different PA molecular species can affect target proteins in a non-identical way leading to specific physiological consequences. Different PA molecular species bind to proteins acting in different branches of ABA responses with dissimilar affinity: di18:1-PA has been shown to be more active in binding protein phosphatase 2C *At*ABI1 than di16:0-PA, di18:0-PA and di18:2-PA [32]; 16:0/18:2-PA binds the NADPH oxidase subunit *At*RbohD preferentially to 16:0/16:0-PA and 18:0/18:0-PA [30]; the binding of 18:1/18:1-PA, 16:0/18:1-PA and 16:0/18:2-PA to the sphingosine kinases *At*SPHK1 and *At*SPHK2 is higher than that of 16:0/16:0-PA [33]. 

The concentration of PA significantly differs depending on plant tissue type, plant developmental phase, treatment condition as well as registration technique. For example, in ^32^P-labelled *Arabidopsis*, basal PA levels range from 2% (in germinating seeds and seedlings) to nearly 8% (in leaf peels) of all labeled phospholipids [25]. In the roots of soybean seedlings, the level of PA, as detected by MS, was found to constitute up to 2.5% of total cell lipids [34]. The level of PA in *Arabidopsis* seedlings detected by GC was found to constitute several mol% [35]. In rice leaves, the level of PA analyzed by mass spectrometry was observed to be in the range of 2.5 to 5 nmol/mg [36]. 

The general topology of PA signaling in plants seems to be similar to that of animals. Plasma membrane or intracellular receptors activate PA production by directly or indirectly affecting the activity of PA-producing enzymes; then, the formed PA binds target proteins and/or affects membrane conformation. However, some differences exist between PA signaling in animals and plants. The first important difference concerns the interrelationships between DAG and PA. For example, in animals, DAG has several protein targets in comparison with a huge number of PA targets and a broader array of signaling properties that are very distinct from those of DAG. On the other hand, in plants, no DAG targets were found, suggesting that PA is the lipid signal instead of DAG. Although conversion of DAG to PA by the DGK is the major route to terminate DAG signaling in animals, there are several examples indicating that DGKs modulate signaling events by producing PA [37]. The second difference is associated with enzymes of PA production. No NPC gene was found in animals in comparison to plants. Furthermore, the PLD family in plants is more heterogenic in comparison with classical PLDs (PLD1 and PLD2) in animals, and PLD substrate specificity is wider in plants than that in animals [38]. The third difference in PA signaling between animals and plants concerns the pathways of PA metabolic conversions. PA phosphatases from animals can generate DAG from PA produced by PLD, and this newly formed DAG possesses signaling and metabolic function [39]. However, such pathways in plants produce DAG for the synthesis of other lipids (e.g., triacylglycerols, galactolipids), but the role of the pathway in signaling was not discovered in plants. In addition, diacylglycerol pyrophosphate, a phosphorylated form of PA with signaling functions, has been found in plants, but not animals [40]. Such a multifaced role of PA in living organisms suggests an evolutionary significance of its signaling mechanisms.

## 2. Spatiotemporal Characteristics of Phosphatidic Acid as a Signaling Molecule

PA has several properties underlying its signaling functions. PA has a specific pattern of intracellular localization. Under basal conditions, PA has been detected in the plasma membrane (PM) in various plant objects: growing tobacco pollen tubes [41,42], mature trichomes [43], epidermal and meristematic root cells and shoot tissues [44]. However, PA has also been detected in cell membranes other than PM. PA has been detected in the nucleus of *Arabidopsis* epidermal root cells undergoing cell division [44] and in chloroplast membranes [45]. Interestingly, the level of nuclear PA in *Arabidopsis* seedlings has been shown to be affected by the activities of lipid signaling enzymes located at the PM, cytosol and endoplasmic reticulum [3]. The level of PA in the PM is highly dynamic even under basal conditions, possibly due to the combination of rapid lateral diffusion of PA and PLD, DGK and LPP activities leading to local accumulation of PA [41]. In response to extracellular stimuli (e.g., abscisic acid), rapid changes in PA signaling, specifically at the PM of plant cells, have been reported [46]. 

Another important property of PA is its high turnover rate. The rapid turnover of PA contributes to its low level in unstressed cells compared to structural phospholipids. The increase in PA content in response to a stimulus is usually transient [46]. 

Cells of different organs and tissue types have different capacities to respond to extracellular stimuli. At least in part, this could be due to differences in their basal PA levels. These levels vary depending on the growth stage of the plant [47], type of plant organ [48,49] or developmental phase of the organ [50]. This could explain why the stimulated PA levels were different in plant organs [51] and tissue types [46,52]. The differential capacity for PA signaling at different developmental stages can be illustrated by the high levels of DGK and PLD proteins in developing maize ears [53] and the high basal PA content in cotton fiber cells compared to ovule cells [54]. The expression levels of many PA signal transduction enzymes also differ depending on the plant developmental phase (Figure 2). These differences may thus determine the extent of activation of lipid signal transduction in response to extracellular (or endogenous) stimuli and control its regulatory performance.

## 3. Effect of PA on Membrane Status

Signaling PA is produced rapidly, transiently and at low concentrations by different pathways in cellular membranes. PA regulatory influence on metabolism is mediated by interactions with target proteins and by inducing effects on membrane functions, charge and conformations.

Phospholipids and enzymes of phospholipid turnover are involved in the control of membrane trafficking [56,57] and the transduction of specific extracellular signals in plants [58,59]. It becomes clear that plant hormone-induced PA accumulation might also be associated with membrane trafficking pathways. This can be illustrated by the fact that *n*-butanol, an inhibitor of PA formation by PLD, impairs translocation of NON-EXPRESSOR OF PATHOGENESIS-RELATED GENES 1 (NPR1) to the nucleus in response to salicylic acid [60].

Endocytosis and exocytosis induced by PA may be an important part of PA signaling in plants. For example, activation of *Brassica napus* PLDα1 during compatible pollination has been hypothesized to promote exocytosis required for pollen germination through the production of PA in the PM [61]. This notion is also supported by genetic and pharmacological data suggesting that the production of PA modulates basal cellular membrane trafficking. Suppression of PA production by inhibition of PLD or DGK activity compromised membrane trafficking, disrupted the deposition of cell wall material (pectins) at the pollen tube tip and inhibited pollen tube growth [62]. Inhibition of PLD also blocked the membrane localization of clathrin and affected the localization of PIN2 (an auxin transporter) in root epidermal cells [63]. In *dgk4 Arabidopsis* plants, a reduced rate of membrane secretion and recycling, as well as inhibited pollen tube growth, were observed [64]. PA and lysophosphatidic acid acyltransferases (AtLPAAT4 and AtLPAAT5) that also generate PA were suggested to be critical for the secretory pathway involved in protein trafficking of the auxin transporter PIN2 and the aquaporin PIP2;7 [65].

A recent study on the functions of PLDδ-family phospholipase in tobacco showed that pollen tubes support a high level of expression of PLDδ genes during growth. For some of the PLDδ isoforms, the unique localization was detected, and the sequences important for binding to the PM were determined [42]. It has been noted that the increased level of PA, formed by the overexpression of *Nt*PLDδ3 isoform, causes specific morphological phenotypes indicating a violation of the balance of vesicular traffic and membrane recirculation [42].

PA and the enzymes that produce it can affect membrane trafficking and endocytosis by interacting with their specific proteins. *At*PLDα1 is associated with both microtubules and clathrin-coated vesicles [66], *At*ECA2 and ANTH domain proteins that recruit clathrin to membranes to facilitate vesicle budding [67], and CHC (Clathrin Heavy Chain) proteins [68]. Other membrane trafficking events regulated by PA include PA-mediated membrane recruitment of *Nt*EXO70A1a and *Nt*EXO70B1—two proteins involved in targeting the vesicle tethering complex (exocyst) for exocytosis [69]. 

Some of the effects of PA on membrane conformations are local: the accumulation of PA leads to a negative spontaneous curvature that facilitates interactions with specific proteins. Indeed, the binding of the Epsin-like Clathrin Adaptor 1 (*At*ECA1) to PA was dependent on membrane curvature stress [70]. Hence, binding was achieved only after a certain threshold level of negative curvature stress. This suggests that the binding of *At*ECA1 requires some degree of hydrophobic interaction with the membrane. PA-binding proteins may take advantage of the presence of bulky hydrophobic residues such as tryptophan to insert themselves into the membrane interface, having a negative spontaneous curvature [71].

Since PA is a component of cell membranes, it interacts not only with proteins but also with other membrane lipids. Studies performed on artificial membranes show that PA binds to phosphatidylcholine (via electrostatic interactions) [72] and stacks with other PA molecules (at low pH or in the presence of divalent cations) [73]. These interactions could be a factor that regulates the affinity of PA towards protein binding and membrane tethering. 

PA signaling could be functionally linked to pH changes. The protonation state of PA has been found to depend on the local pH and to affect the efficiency of PA binding to its effector proteins [46]. 

## 4. Binding of Phosphatidic Acid to Target Proteins

A number of PA-binding proteins have now been identified. Many of them are thought to play a role in plant signal transduction. However, a detailed characterization of the biological role, as well as the deciphering of the interaction with PA at the molecular level, is available for only some of them (Table 1). Since no PA-recognition motif has yet been identified in plant proteins (as well as animal proteins), it has been proposed that complex structural folds of a protein are involved in PA recognition. PA-binding modules typically consist of an amphipathic α-helix with positively charged amino acids on one side (e.g., lysine-rich sequence motif) and hydrophobic residues on the other [74]. Similar structural folds have been found in proteins involved in plant hormone signaling, suggesting that this may indeed be a common mechanism of PA-protein interaction (Figure 3). 

An interaction between a PA and a protein implicates specific amino acid residues of the latter. It has been shown that positively charged polybasic residues (Lys, Arg) [30,75] are involved in PA binding. The negative charges carried by anionic PA thus enable the recruitment of proteins with polybasic regions [44]. In addition, other types of amino acids (e.g., His in the defensin *Hs*AFP1 [76] and Pro in the lipid kinase *At*PIPK1 [77]) are suggested to act in the binding of PA. Together, this enables the establishment of electrostatic and hydrophobic interactions as well as hydrogen bonds between PA and a target protein. His, Trp, Ser, Ala and Gly are also commonly present in PA-binding modules of proteins. This is, for example, true for CDeT11-24, a desiccation-related phosphoprotein of *Craterostigma plantagineum*. The Lys-rich motif of this protein is responsible for the interaction with the PA phosphomonoester moiety, while a region of nonpolar amino acids, seen in a helical wheel projection of the lysine-rich motif of CDeT11-24, may be involved in hydrophobic interactions with the acyl chain of PA [78]. His42 and His51 are thought to contribute to the pH sensitivity of the binding of PA to the Epsin-like Clathrin Adaptor 1 [71]. Non-basic amino acids could play a supporting role in PA binding and cause a change in the PA binding affinity of a protein—this could be an explanation for the differences in PA binding to the NADPH oxidases *At*RbohD and *At*RbohF [30].

Interestingly, several PA-binding sites are sometimes located in different parts of a protein. It should be noted that multiple basic and hydrophobic amino acid residues have been shown to be important elements of the *At*PINOID kinase binding to PA [79]. *At*SnRK2.4 contains multiple basic amino acid residues located in different regions, all involved in PA-binding [80]. In the case of the potassium channel *Os*AKT2, PA was found to bind two adjacent lysine residues in the cytosolic ANK domain [81] and basic residues in the S4 voltage-sensing domain. The cytosol-facing domains of *Os*AKT2 contribute to the PA-mediated negative regulation of *Os*AKT2 activity [82]. In addition, the EH1.1 domain of the TPC subunit *At*EH1/Pan1 (adaptor complex of clathrin-mediated endocytosis machinery) coordinates the PA molecules by three main regions (residues around K42, residues around K78 and residues located at the C-terminus around K97), whereas the EH1.2 domain interacts with the PA molecules at the region forming the phosphoinositide-binding site and at the region located close to the protein C-terminus. In contrast to EH1.1, EH1.2 binds a higher number of PA molecules [83]. 

The different affinities of PA species towards binding to target proteins may act as important determinants of PA signal specificity in response to dissimilar stimuli. The role of fatty acid tails in altering the affinity of PA towards a protein is likely determined by the specific structural features of the letter: e.g., the presence of a stretch of hydrophobic residues in the PA-binding region of protein phosphatase 2C *At*ABI1 and protein kinase *At*CTR1 [32,84]; presence of hydrophobic residues at the C-terminus of *At*CP, an actin-capping protein [85]; hydrophobic cavity in the structure of *At*SPHK1 [86]. PA is a cone-shaped molecule that induces negative spontaneous curvature [87], facilitating “deeper” insertion of proteins into lipid membranes, providing a background for the role of hydrophobic interactions [70]. On the other hand, some proteins, e.g., an MYB transcription factor (*At*WER) [88] and glyceraldehyde-3-phosphate dehydrogenases (*At*GAPC1 and *At*GAPC2) [89] do not show specific preferences for binding to particular PA species. For these proteins, binding is likely determined by the interaction of the phosphomonoester group of PA with basic residues.
Figure 3PA-binding folds of *Arabidopsis* proteins involved in plant hormone (abscisic acid) signaling. Shown are amphipathic α-helix projections of a minimal characteristic PA-binding domain obtained with “Heliquest” software [90]. The arrows indicate the hydrophobic moment. Basic and hydrophobic residues are shown in blue and yellow, respectively. PA-binding motifs were obtained from published data: AtRbohD [30], AtABI1 [32], AtRGS1 [91], AtGEF8 [75] and AtSPHK1 [86]. The sequences of PA-binding motifs are as follows: 140-SRELRRVFSRRPSPAVRRFD-159 (AtRbohD), 63-GSESRKVLISRINSPNLNMK-82 (AtABI), 250-QPLLSQISLKKRQNFEFQRM-269 (AtRGS1), 6-ERGLSASKSFNFKRMFDSSS-25 (AtGEF8), 175-KYDGIVCVSGDGILVEVVNG-194 (AtSPHK1).
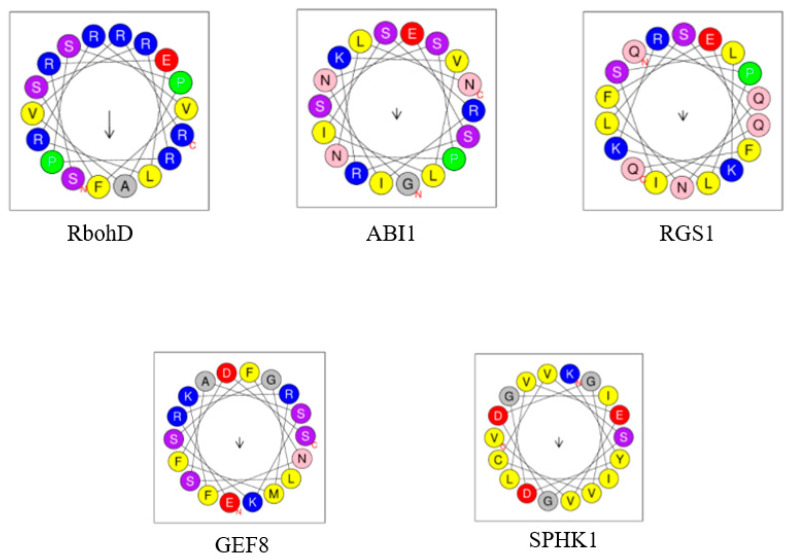



Binding to PA is not necessarily associated with a tethering of proteins to PM. It has been shown that the binding of the cytosolic glyceraldehyde-3-phosphate dehydrogenase *At*GAPC2 to PA induces proteolytic cleavage of the enzyme [89]. It was found that PA interacts with the transcription factors *At*AHL4 (AT-hook motif-containing nuclear localized 4) [3] and *Arabidopsis* LATE ELONGATED HYPOCOTYL/CIRCADIAN CLOCK ASSOCIATED1 [92] and negatively regulates their DNA-binding ability. For *At*AHL4, PA-mediated tethering to the nuclear membrane was excluded as a mechanism of PA action, suggesting a hypothetical possible association of PA with *At*AHL4 beyond membranes [3]. PA also binds to the gibberellin receptor *Os*GID1, promotes its nuclear localization, and mediates the gibberellin response in rice [36]. 

PA could regulate cell signaling by simultaneous binding to multiple components of the same regulatory pathway. One example is the binding of PA to the potassium channel *At*AKT2 [81] and *At*PP2CA (protein phosphatase 2CA, negative modulator of *At*AKT2 activity) [93]. Similarly, PA binds to *At*MKK7/*At*MKK9, which phosphorylate PA-binding *At*MPK6 [94]; activates *At*MKK5, and its downstream targets *At*MPK3/*At*MPK6 [95]; binds *At*PINOID kinase [79], and *Arabidopsis* D6 PROTEIN KINASE [96], all of which regulate PIN auxin transporter activity within the same pathway. PA has been shown to regulate not only the activity, phosphorylation status and/or localization of *At*SnRK2, but also that of its cellular partners: *At*SnRK2 enzyme targets (e.g., *At*ERD14) and negative regulators (*At*ABI1 and *At*SnRK2-interacting calcium sensor) [97]. 

## 5. Role of PA as a Lipid Second Messenger of Signal Transduction Pathways in Plants

### 5.1. Phosphatidic Acid Signaling in Hormonal Regulation

PA plays a key role in the regulation of metabolism in plants as well as in other living systems. PA is a mediator in the signal transduction pathways of hormones and other biologically active compounds that ensure plant growth, development and resistance to adverse environmental factors. Regulatory signals originating from hormones such as auxin, brassinosteroids, salicylic acid, abscisic acid, jasmonic acid and gibberellins have been found to implicate PA as a second messenger. PA, which is produced in response to the stimuli, has been found to be involved in genomic and non-genomic signaling events that regulate gene expression, ion transport, cytoskeletal dynamics and metabolic enzyme activities (Figure 4).

#### 5.1.1. Role of Phosphatidic Acid in Auxin Signaling

Genetic studies suggest that the phospholipases AtPI-PLC2 [98] and AtNPC5 [99] play a role in auxin sensing pathways. PA binds to specific signaling proteins involved in the regulation of auxin transport and regulates their activity. For example, PA-binding 3’-phosphoinositide-dependent protein kinase1 (AtPDK1) phosphorylates and regulates PA-binding D6 protein kinase (AtD6PK) to stimulate the activity of PIN auxin efflux carriers [100]. PLD and PA were suggested to be involved in auxin-mediated reversion of the actin bundling induced by the hairpin elicitor [101]. 

#### 5.1.2. Role of Phosphatidic Acid in Salicylic Acid Signaling

Salicylic acid (SA) activates PA production via DGK [102] and PLD [103,104,105,106] in *Arabidopsis thaliana* and *Capsicum chinense*. PLD and PA have been shown to be involved in transcriptome regulation by SA [105,107,108], SA-induced ROS accumulation via the NADPH oxidase AtRbohD [103], SA-induced metabolic changes [109] and nuclear translocation of AtNPR1, an SA signaling regulator [60]. 

#### 5.1.3. Role of Phosphatidic Acid in Brassinosteroid Signaling

Brassinosteroids (BR) trigger the accumulation of DAG and PA derived from phosphatidylcholine in various plant tissues due to the activation of NPC and DGK [110,111,112]. AtNPC3 and AtNPC4 isozymes are involved in plant sensitivity to BR in terms of root growth and gene expression [110]. Several DGK isoforms contribute to the production of PA in response to BR. Epibrassinolide (EBL) treatment resulted in a two-fold increase in PA in WT *Arabidopsis thaliana* roots [113]. However, a similar response was reduced in *dgk1dgk2*, *dgk1* and *dgk5dgk6* mutant lines. Of all the *dgk* mutant lines tested, *dgk3* plants were characterized by the lowest PA induction in response to EBL treatment compared with WT plants. Pretreatment of plants with BR further increased the levels of DAG and PA during subsequent salt stress. In this context, it has been suggested that DAG and PA are involved in the BR-induced adaptation of plants to high salinity [112]. Moreover, a reduced EBL-stimulated PA production in *dgk* mutants might be linked to lower germination rates, lower total and alternative respiration under optimal and high salinity conditions [113]. 

#### 5.1.4. Role of Phosphatidic Acid in Abscisic Acid Signaling

Abscisic acid (ABA) is one of the most important plant hormones with respect to the regulation of plant metabolism (e.g., acting in seed germination, growth responses) and stress resistance (e.g., by controlling stomatal closure). ABA-induced rapid PA generation has been localized to the PM of *Arabidopsis thaliana* guard cells and cells of the root maturation zone [46]. PA plays a role as a second messenger in ABA-regulated events such as stomata closure and water stress tolerance. Specific PA-binding proteins are known to be involved in ABA signaling in plants. ABA-activated PLD leads to PA production, which inhibits AtRGS1—a regulator of G-protein signaling [91] and activates the guanine nucleotide exchange factor AtGEF8 [75]. Moreover, recruitment of ABI1 to the PM by the leucine-rich receptor-like kinase AtRDK1 in response to ABA requires the binding of PA to the former [114]. PLD and PA are also involved in ABA-induced microtubule depolymerization and calcium spikes in guard cells [115] and HVA1 expression [116].

The functional role of PA in ABA-signaling pathways could be illustrated by in silico analyses, which showed similarities in the gene expression patterns of ABA-treated wild-type plants and untreated *rgs1* lines. It is known that *At*RGS1 is inhibited by PA as part of the ABA-signaling pathway [84]. Therefore, this signaling event is mimicked in *rgs1* plants, at least in part, by the absence of ABA. Comparing the two sets of gene expression data, we found that a number of genes encoding signaling, metabolic or stress-related proteins are regulated in a similar manner (Figure 5). These data suggest that inhibition of *At*RGS1 by PA is indeed an important event in the ABA response in *Arabidopsis*.

#### 5.1.5. Role of Phosphatidic Acid in Jasmonic Acid Signaling

Jasmonic acid is a hormone that rapidly activates the production of PA via PI-PLC/DGK and PLD in potato tubers [119], leaves of *Brassica napus* [120] and cells of *C. chinense* [102,121]. Jasmonate-induced accumulation of the secondary metabolites [102] and potato tuber cellular growth [119] were found to be downstream of PI-PLC\PLD activity and PA accumulation. 

#### 5.1.6. Role of Phosphatidic Acid in Gibberellin Signaling

Gibberellins were also suggested to interact with PA signaling. In rice, among several PLD isoforms tested, only OsPLDα6 was found to be an important positive regulator of gibberellin sensitivity in terms of modulation of growth responses. Results from lipidomic analysis indicate that phosphatidylethanolamine and phosphatidylserine are the main sources of PA in response to gibberellin. In rice protoplasts, gibberellin induces OsPLDα6 translocation from cytosol to nuclei. PA formed by OsPLDα6 activity positively mediates gibberellin signaling via PA binding to gibberellin receptor OsGID1, promotion of its nuclear localization and degradation of the DELLA protein SLENDER RICE1, a negative regulator of the hormone signaling [36]. In contrast, the level of suppressor protein SLENDER RICE1 was decreased in npc6 rice mutants treated with gibberellin, but drastically increased in OsNPC6 over-expressing plants [122]. Therefore, OsPLDα6 and PA are positive players of early events of gibberellin signaling in plants. 

Hormones can also modulate PA signaling by controlling gene expression of PA-producing enzymes (Figure 6). Although the enzymes of PA turnover are well known, controlling the expression level of their encoding genes could still have a significant impact on the resulting signaling function. Thus, downregulation of *AtNPC6* expression following treatment with ABA could be a mechanism that attenuates the signaling response (Figure 6). Conversely, upregulation of *AtPLDδ*, *AtPLDζ2* and *AtDGK1* by the same stimuli could mean that the abundance of these isoenzymes is important to fulfill the latter steps in the ABA signaling pathway. On the other hand, cytokinins weakly affect PLD gene expression (Figure 6), but the role of PLD and PA as mediators of cytokinin-regulated gene expression was found [123,124].

### 5.2. Phosphatidic Acid as a Signal Transducer for Other Biologically Active Substances

Some hormone-like substances also trigger PA signaling. It has been shown that treatment with polyamines activates *At*PLDα1 [125], *At*PLDδ and DGK [52] in *Arabidopsis* seedlings and also PLD and DGK in *C. arabica* cells [126], leading to a rapid PA production. This response was dependent on the type of polyamines tested and their charge. For example, putrescine triggered higher *At*PLDα1 activation in vitro compared to spermine and spermidine [125], while in *Arabidopsis* seedlings, spermine, thermospermine and spermidine induced higher PA accumulation compared to putrescine and diaminopropane [52]. Cellular uptake of spermine by the PM localized amino acid transporter Resistant to Methyl Viologen 1 (*At*RMV1) is required for triggering PA production [52]. As a port of cell response to polyamines, *At*PLDα1-produced PA was shown to trigger H_2_O_2_ accumulation in guard cells [125], while *At*PLDδ-produced PA triggers potassium ion efflux in the root elongation zone of *Arabidopsis* [53].

SCOOP12—an active endogenous plant peptide (phytocytokine)—has been shown to rapidly stimulate the production of PA via the PI-PLC/DGK pathway in *Arabidopsis* cells [127]. During the self-incompatibility response of *Brassica napus* triggered by the interaction of SP11 peptide with SRK (S-locus receptor kinase), PLDα1 was found to be a target of ARC1-mediated ubiquitination. Subsequent degradation of PLDα1 thus impedes the production of PA, which itself is required for pollination. It has been proposed that activation of the PLDα1 isoform by an unknown ligand during compatible pollination would lead to PM remodeling supported by the production of PA to promote exocytosis required for pollen germination [61]. Other data indirectly suggest a possible role of PA pathway enzymes in peptide signaling. *At*PI-PLC2 has been found to interact with *At*PXL1—a receptor for the peptide *At*CLE41 [128]. Moreover, some peptide receptors might be co-expressed with PA-producing enzymes, as suggested by in silico analysis (see Table 2). This could indicate a possible functional link between certain peptide receptors and a signal lipid production.

PA signaling may also interact with signaling pathways of other bioactive substances. Both *At*PLDα1 and *At*PLDδ are involved in hydrogen sulfide signaling leading to stomatal closure in *Arabidopsis*. Hydrogen sulfide-induced H_2_O_2_ accumulation in guard cells was specifically dependent on *At*PLDα1 but not on *At*PLDδ. In contrast, hydrogen sulfide treatment led to PA accumulation in an *At*PLDδ-dependent manner. Interestingly, hydrogen sulfide in *rbohD* lines did not induce PA accumulation in guard cells [129]. RbohD is an enzyme known to be regulated by PA [30], but in hydrogen sulfide signaling it, apparently, acts upstream of PLD-generated PA.

Extracellular *Pbr*S-RNase, which is a proteinaceous female determinant of pear self-incompatibility, induces rapid PA accumulation that depends on *Pb*rPLDδ1 expression in pollen tubes. PA thus prevents depolymerization of the actin cytoskeleton induced by *Pbr*S-RNase and delays self-incompatibility signaling [130].

Biologically active substances can also modulate PA signaling by controlling gene expression of PA-producing enzymes (Figure 6). Consistent with this, the complex regulation of multiple PI-PLC and NPC genes by melatonin in soybean [131] supports the notion that bioregulators define an appropriate expression pattern of lipid signal transduction enzymes.
Figure 6Meta-analysis of the expression pattern of PA-producing signaling enzymes (PLD, PI-PLC, NPC, DGK) in plants in response to phytohormones. Transcriptomic data were collected from the “Genevestigator” database [55] according to the outputs of classical array experiments (“AT_AFFY_ATH1” platform). Shown here is a heat-map of the relative values of gene expression differences between hormone-treated and mock-treated samples. All statistical data were implemented and provided by the “Genevestigator” database. Color saturation corresponds to the specific level of upregulation (red) and downregulation (green) of gene expression. Genes and corresponding AGI used: *At*PLDα1 (At3g15730), *At*PLDα2 (At1g52570), *At*PLDα3 (At5g25370), *At*PLDε (At1g55180), *At*PLDβ1 (At2g42010), *At*PLDβ2 (At4g00240), *At*PLDγ1 (At4g11850), *At*PLDγ2 (At4g11830), *At*PLDγ3 (At4g11840), *At*PLDδ (At4g35790), *At*PLDζ1 (At3g16785), *At*PLDζ2 (At3g05630); *At*PI-PLC1 (At5g58670), *At*PI-PLC2 (At3g08510), *At*PI-PLC3 (At4g38530), *At*PI-PLC4 (At5g58700), *At*PI-PLC5 (At5g58690), *At*PI-PLC7 (At3g55940), *At*PI-PLC8 (At3g47290), *At*PI-PLC9 (At3g47220), *At*PI-PLC10 (At4G34920), *At*PI-PLC11 (At3G19310); *At*NPC1 (At1g07230), *At*NPC2 (At2g26870), *At*NPC3 (At3g03520), *At*NPC4 (At3g03530), *At*NPC5 (At3g03540), *At*NPC6 (At3g48610); *At*DGK1 (At5g07920), *At*DGK2 (At5g63770), *At*DGK3 (At2g18730), *At*DGK4 (At5g57690), *At*DGK5 (At2g20900), *At*DGK6 (At4g28130), *At*DGK7 (At4g30340). Gene of *At*PI-PLC6 (At2g40116) was not represented in the corresponding platform. A total of 15 experiments were used for the analysis. Abbreviations, short description of experimental conditions (time of treatment, concentrations, tissue types, genotypes) and experiment ID codes for each set of experimental data (from figure left to right) are as follows: ABA, abscisic acid, 10 μM, 3 h, seedlings, Col-0 (AT-00110); ACC, 1-aminocyclopropane-1-carboxylic acid, 10 μM, 3 h, seedlings, Col-0 (AT-00110); ethylene, 5 ppm, 3 h, petiole samples, Col-0 (AT-00013); BA, 6-benzylaminopurine, 10 μM, 1 h, shoot samples, Col-0 (AT-00351); zeatin, 1 μM, 3 h, seedlings, Col-0 (AT-00110); BL, brassinolide, 10 nM, 3 h, seedlings, Col-0 (AT-00110); 24-eBL, 24-epibrassinolide, 0.1 μM, 3 h, dark, seedling sample, Col (AT-00650); GA, gibberellic acid 3, 1 μM, 3 h, seedlings, Col-0 (AT-00110); IAA, indole-acetic acid, 1 μM, 3 h, seedlings, Col-0 (AT-00110); NAA, naphthaleneacetic acid (a synthetic auxin), 10 μM, 2 h, leaf disc samples, Col-0 (AT-00392); MeJa, methyl jasmonate, 10 μM, 3 h, seedlings, Col-0 (AT-00110); OPDA, 12-oxo-phytodienoic acid, 75 μM, 4 h, seedlings, Col-0 (AT-00293); SA, salicylic acid, 2 mM, 1 day, seedlings, Col-0 (AT-00320); RALF, Rapid Alkalinization Factor 1 (peptide, *At*RALF), 1 μM, 30 min, seedling sample, Col (AT-00679); strigolactone, GR24, 1 μM, 90 min, *max3-9* (a strigolactone deficient mutant) whole plant samples (AT-00404). Original data resources: GSE14961/E-GEOD-14961 datasets for salicylic acid-regulated gene expression (AT-00320), [132,133,134,135,136,137,138,139] for other hormones/biologically active substances.
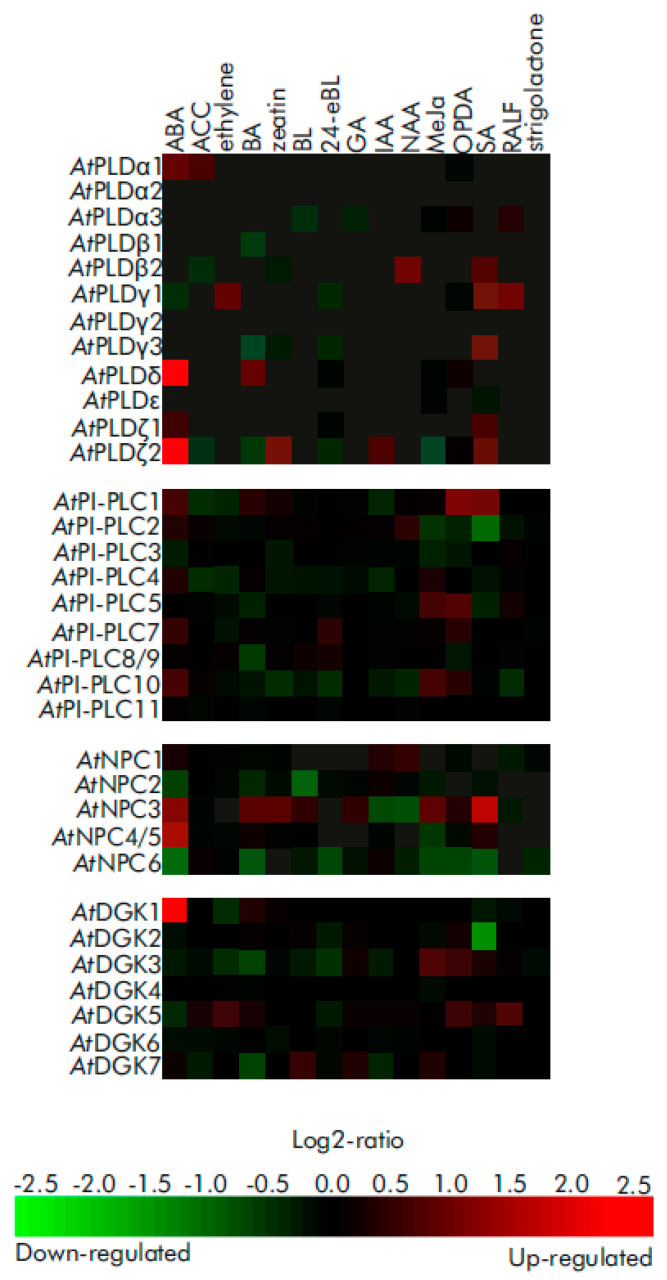



## 6. The Most Common Methods for the Studying of Phosphatidic Acid Signaling in Plants

The detection of transient and minute changes in PA production in plants requires a specific methodological strategy. Current approaches for the measuring of PA content include radioisotope labeling, chromatographic techniques and mass-spectrometric methods [140,141]. PA changes are often rapid, requiring appropriate timing and rigorous handling, extraction and determination of PA, which are time-consuming and expensive.

Recent insights into the nature of protein-phospholipid interactions have enabled the development of genetically-encoded fluorescent molecular probes that can specifically interact with various phospholipids and allow their visualization in living cells. To this end, PA-specific probes containing a PA-binding domain are fused with fluorescent proteins and expressed in plant cells. For example, Potocky et al. [41] constructed a PA biosensor consisting of a Spo20p-PA-binding domain fused with a fluorescent YFP protein that reflects the dynamics of PA in transiently transformed tobacco pollen tubes, as shown by confocal laser scanning microscopy [41]. Subsequently, Li et al. developed a Förster resonance energy transfer technology (FRET)-based PA-specific biosensor PAleon that monitored and visualized the concentration and dynamics of PA in plant cells under abiotic stress [46]. These PA biosensors have helped to reveal the spatiotemporal dynamics of PA in plants. Similarly, DAG, PS, PIP_2_ and PIP lipids biosensors have also been successfully used in plant cells [142]. These biosensors can be used for real-time monitoring of lipid dynamics and providing new insights into lipid metabolism and signaling in plants.

The PA content in plant tissue extracts can also be measured using a PA enzyme-linked immunosorbent assay (ELISA) with an anti-PA antibody, where the PA content is calculated using a standard curve [143]. In addition, a fluorometrically-coupled enzyme assay for measuring PA is also known. It involves PA deacylation by added lipase and further oxidation of PA-derived glycerol-3-phosphate to generate hydrogen peroxide. The latter is subsequently used for the peroxidase-catalyzed conversion of Amplex Red reagent to a fluorescent resorufin. The fluorescence intensity of resorufin is then measured using a fluorescence microplate reader [144].

## 7. Conclusions

Despite the well-documented role of PA as a biologically active molecule, we still lack an understanding of the underlying mechanisms. In particular, it is still not clear how the enzymes of PA production in the PM are rapidly activated (usually within minutes) in response to hormones. Not all hormone receptors are located in the PM, so a spatial relationship can be inferred; subsequent studies will be required to produce a more complete outlook.

Deciphering the role of PA in the regulation of multiple cellular activities remains to be a challenge for future research. For example, under nutrient limitation, cells induce autophagy that delivers cellular cargoes for degradation as a part of the nutrient recycling mechanism. The role of PA in plant autophagy could be inferred from data on plant [145] and animal [146] systems. In *Arabidopsis*, *At*PLDε has been found to promote autophagy in plant response to nitrogen deficiency. Physiological analysis indicates that *At*PLDε acts to delay senescence in response to nitrogen deficiency. The transcript level of *At*PLDε was increased in these stress conditions, while analysis of *At*PLDε-GFP cellular distribution indicated that nitrate starvation increased protein levels of *At*PLDε [145] and promoted its association with intracellular membranes. Co-immunoprecipitation and bimolecular fluorescence complementation analysis indicate that *At*PLDε interacts with *At*ATG8, an autophagy-related protein that is essential for the biogenesis, cargo recruitment and maturation of autophagosomes. Phosphatidylethanolamine (PE) lipidation of *At*ATG8 is necessary to its functions; an incubation of affinity-purified protein *At*PLDε with membrane proteins in plants expressing GFP-*At*ATG8 indicated that *At*PLDε delipidates *At*ATG8-PE complex. Results of immunoblotting suggested that during nitrogen starvation, *At*PLDε increases the abundance of the *At*ATG8-PE complex. Genetic analysis of GFP-AtATG8-labeled *At*PLDε-mutated plants demonstrated that *At*PLDε promotes autophagosome formation in response to nitrogen deprivation. Monitoring of the levels of *At*ATG1a and *At*NBR1 (NEIGHBOR OF BRCA1) proteins that are known to be degraded by autophagy under stress suggested that *At*PLDε increases their autophagic degradation under nitrogen deficiency [145]. Therefore, PLD and PA could play a multifaced role in plant autophagy.

The mechanisms and specificity of binding of PA to proteins are also not fully understood. Despite a proposed mechanism primarily involving a phosphomonoester group of PA (hydrogen bond switch model), a role for the variable fatty acid tails of PA in protein binding is apparent. More experimental evidence needs to be collected before the full PA-binding properties of proteins can be considered.

Another unsolved problem is to link the signaling function of PA with a metabolic one. PA is a critical intermediate in lipid turnover and plays a role in recycling orthophosphate into sink tissues, thus minimizing phosphate starvation [147,148]. Furthermore, a basal PA deficiency in *pldα1* apparently leads to altered abundances of proteins involved in various housekeeping activities [149,150].

Moreover, not only the increase in PA content might have a regulatory effect. The rapid decrease of basal PA by abscisic acid and salicylic acid in *Arabidopsis* cells [107] raises the question of whether it also has a signaling function.

An intriguing fact is the PA’s ability to bind transcription factors and negatively regulate their DNA binding [3,92]. This suggests that the abundance of PA in the nuclear membrane may be a factor controlling the transcriptome through a molecular interface yet to be identified.

## Figures and Tables

**Figure 1 ijms-23-03227-f001:**
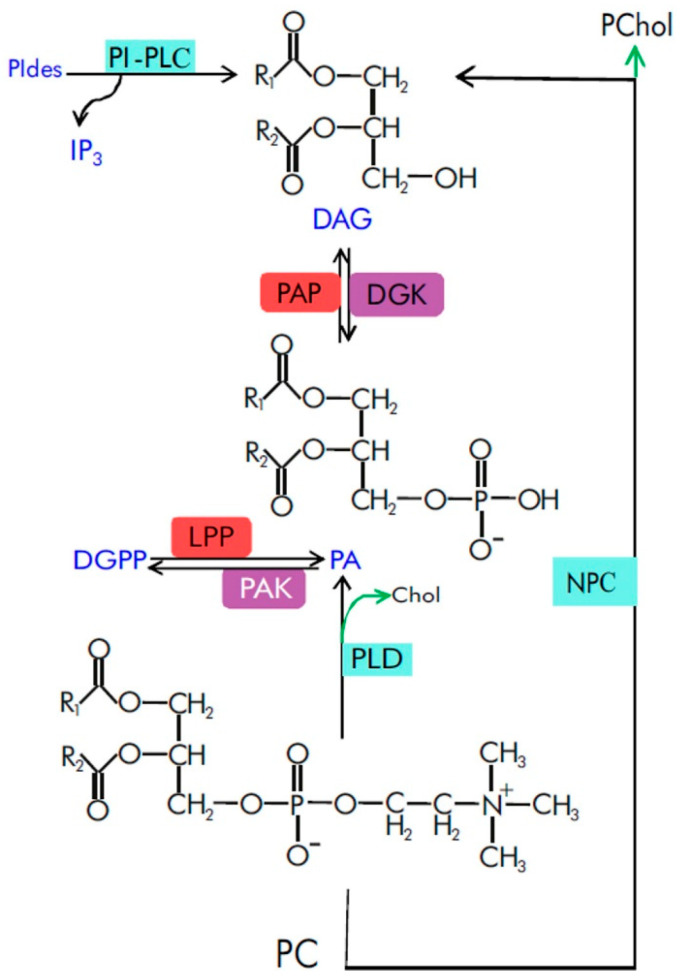
A diagram representing key enzymatic steps of PA production involved in cell signaling. Phosphatidylcholine (PC) is shown as the principal substrate for PLD-generated PA. Signaling lipid mediators and their active products are highlighted in dark blue. Their non-signaling by-products or lipid precursors are shown in black. Phospholipases involved in DAG and PA metabolism are shown in light blue rectangles. Lipid kinases that phosphorylate DAG and PA are shown in violet rectangles. Lipid phosphatases that dephosphorylate DGPP and PA are shown in light red rectangles. Black arrows indicate pathways that generate signaling lipids; green arrows indicate pathways that generate by-products. Note that no PAK-coding gene is currently identified in *Arabidopsis*. Chol, choline; DGK, diacylglycerol kinase; DGPP, diacylglycerol pyrophosphate; IP_3_, inositol 1,4,5-trisphosphate; LPP, lipid phosphate phosphatase; NPC, nonspecific phospholipase C; PAK, phosphatidic acid kinase; PAP, phosphatidic acid phosphatase; PChol, phosphocholine; PC, phosphatidylcholine; PIdes, phosphoinositides, PI-PLC, phosphatidylinositol-specific phospholipase C; PLD, phospholipase D; R_1_, R_2_, fatty acids.

**Figure 2 ijms-23-03227-f002:**
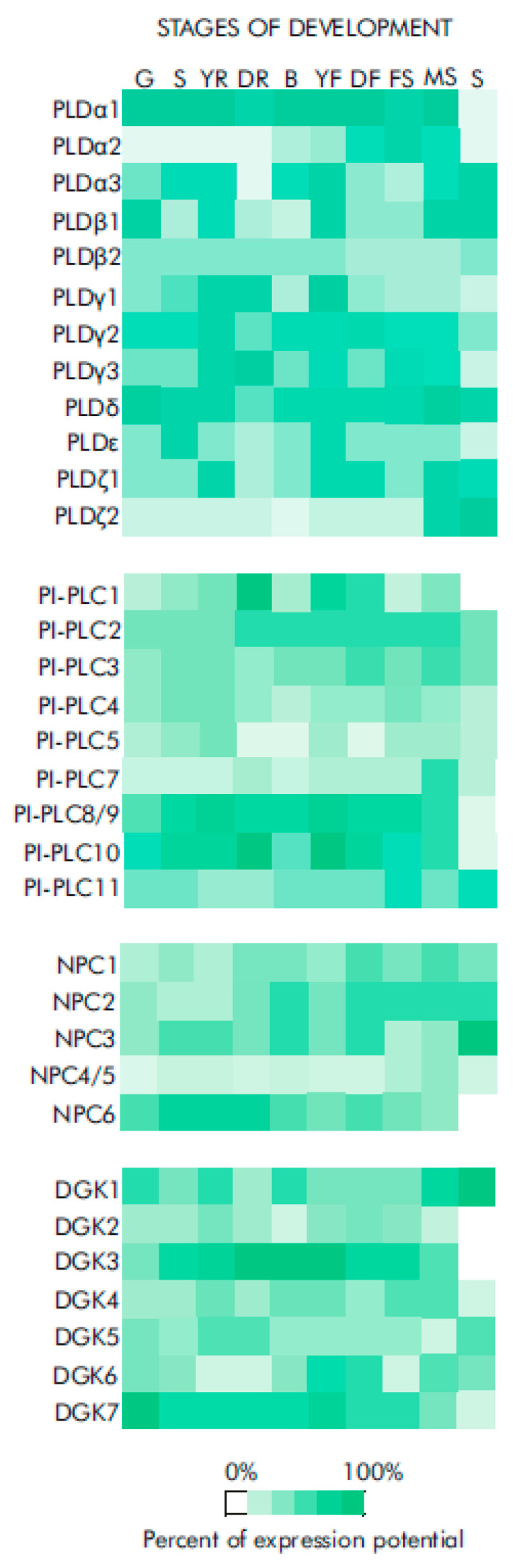
Expression levels of genes encoding enzymes of PA signaling at different developmental stages of *Arabidopsis thaliana* obtained using “Genevestigator” database [55]. Data were collected from data based on classical array experiments (“AT_AFFY_ATH1” platform). The relative values of gene expression (percent of expression potential, log2 scale) are shown. Percent of expression potential denotes the relative transcript abundance given as percentage of the level in the developmental stage where the gene is expressed at maximum level. Color saturation corresponds to the specific level of gene expression. White color signifies an absence of data. Number of samples: germinated seed (515), seedling (2781), young rosette (830), developed rosette (2219), bolting (358), young flower (720), developed flower (1003), flowers and siliques (274), mature siliques (93), senescence (18). Gene identifiers are as follows: PLDα1 (At3g15730), PLDα2 (At1g52570), PLDα3 (At5g25370), PLDε (At1g55180), PLDβ1 (At2g42010), PLDβ2 (At4g00240), PLDγ1 (At4g11850), PLDγ2 (At4g11830), PLDγ3 (At4g11840), PLDδ (At4g35790), PLDζ1 (At3g16785), PLDζ2 (At3g05630); PI-PLC1 (At5g58670), PI-PLC2 (At3g08510), PI-PLC3 (At4g38530), PI-PLC4 (At5g58700), PI-PLC5 (At5g58690), PI-PLC6 (At2g40116), PI-PLC7 (At3g55940), PI-PLC8 (At3g47290), PI-PLC9 (At3g47220), PI-PLC10 (At4G34920), PI-PLC11 (At3G19310); NPC1 (At1g07230), NPC2 (At2g26870), NPC3 (At3g03520), NPC4 (At3g03530), NPC5 (At3g03540), NPC6 (At3g48610); DGK1 (At5g07920), DGK2 (At5g63770), DGK3 (At2g18730), DGK4 (At5g57690), DGK5 (At2g20900), DGK6 (At4g28130), DGK7 (At4g30340). Abbreviations of *Arabidopsis* developmental stages: G, germinated seed; S, seedling; YR, young rosette; DR, developed rosette; B, bolting; YF, young flower; DF, developed flower; FS, flowers and siliques; MS, mature siliques; S, senescence.

**Figure 4 ijms-23-03227-f004:**
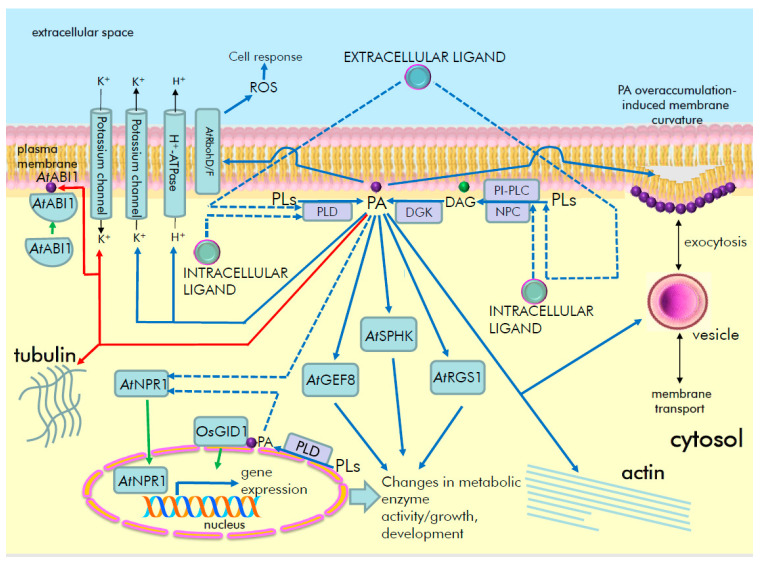
Schematic representation of the PA signaling in plants. Hypothetical receptors lead to activation of lipid signaling enzymes. Key examples of PA-regulated signaling events in hormonal regulation are shown (for details, see the main text); also shown here are PA-induced changes of membrane curvature and subsequent membrane transport events. Headgroups of DAG in a membrane are shown in green; headgroups of PA in crimson; and headgroups of other lipids in pink color. ABI1, ABSCISIC ACID-INSENSITIVE 1, *At*, *Arabidopsis thaliana*, DAG, diacylglycerol, DGK, diacylglycerol kinase, GEF8, RopGEF8, guanine nucleotide exchange factor 8, GID, GIBBERELLIN INSENSITIVE DWARF1, NPC, non-specific phospholipase C, NPR1, NONEXPRESSOR OF PATHOGENESIS-RELATED GENES 1, *Os*, *Oryza sativa*, PA, phosphatidic acid, PI-PLC, phosphatidylinositol-specific phospholipase C; PLD, phospholipase D, PLs, phospholipids (e.g., phosphatidylcholine, phosphatidylethanolamine), RbohD/F, respiratory burst oxidase protein D/F, RGS1, Regulator of G protein Signaling 1, ROS, reactive oxygen species, SPHK, sphingosine kinase.

**Figure 5 ijms-23-03227-f005:**
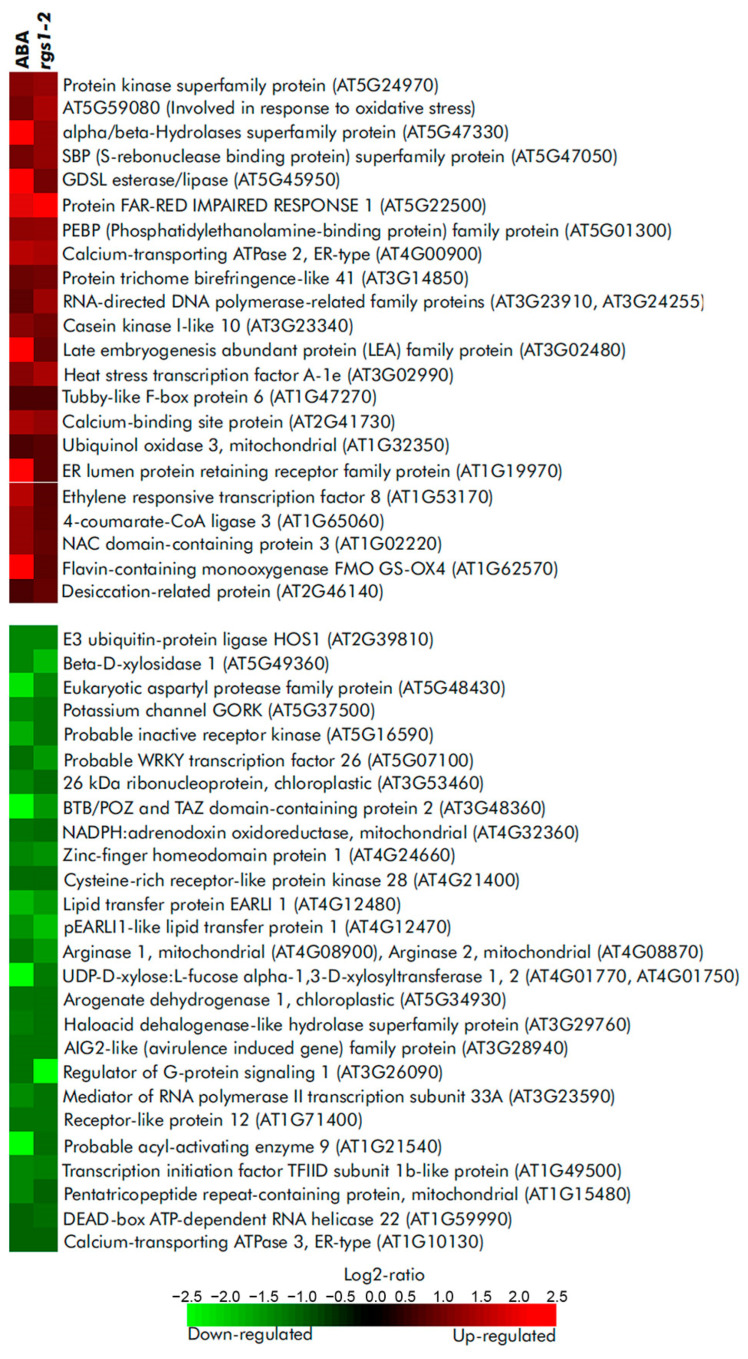
A set of similarly regulated genes in ABA-treated Col-0 *Arabidopsis* plants and in mock-treated *rgs1-2* knockout mutants obtained using “Genevestigator” database [55]. Experiments ID used to mine the data are as follows: ABA, AT-00433; *rgs1-2*, AT-00530. Changes in gene expression (log2-ratio) are color-coded according to the provided scale. Statistical treatment of the data was implemented by “Genevestigator” database. Original data resources [117,118].

**Table 1 ijms-23-03227-t001:** Plant PA-binding proteins with characterized mechanism of PA binding and identified role in cell signaling.

PA Target	PA-Binding Site	Effect on Activity	Role in Signaling	Lipid Signaling Pathway	Notes (PA Binding Specificity)	Refs.
*At*RbohD (At5g47910), *At*RbohF (At1g64060), respiratory burst oxidase homolog protein D and F	Arg149, Arg150, between EF-hands and N-terminus (in cytosolic region), in the vicinity of phosphorylation motif (*At*RbohD); residues 105 to 341 (*At*RbohF)	Stimulation at PM (*At*RbohD)	ABA signaling, ROS generation, ROS-dependent NO production, stomatal closure	*At*PLDα1	di18:1PA, di18:2PA, 16:0–18:1 PA, 16:0–18:2PA, 18:0–18:2 PA), but not to di16:0PA, di18:0PA; *At*RbohD has higher affinity to PA than *At*RbohF	[60]
*At*SPHK1 (At4g21540), *At*SPHK2 (At4g21534), sphingosine kinases	VSGDGI motif (*At*SPHK1)	Stimulation by promoting substrate binding	ABA signaling, *At*PLDα1 activation (for amplification of ABA response), stomatal closure, seed germination, root elongation	*At*PLDα1	Strong binding to 18:1/18:1PA, 16:0/18:1PA, 16:0/18:2PA	[71,72]
*At*RopGEF8 (At3g24620), guanine nucleotide exchange factor 8	Lys13, Lys18 in *At*GEF8 PRONE domain and N-terminus	Stimulation of *At*GEF8 activity towards small GTPase ROP7	ABA signaling activation		di18:2PA, di18:1PA, 16:0–18:1PA, 16:0–18:2PA, 16:0–18:1PA, 18:0–18:2PA	[61]
*At*RGS1 (At3g26090), regulator of G-protein signaling	Lys259 in RGS domain	Inhibition of GTPase-accelerating protein activity of *At*RGS1	ABA signaling activation, inhibition of seed germination, root growth and stomatal responses	-	-	[73]
*At*MAP65-1 (At5g55230), a microtubule-associated protein	Lys and Arg from 53KRK55 and 61KSR63 sequences, Ser-428 in 428SK429 sequence, out of microtubule binding or phosphorylation regions	Promotion of *At*MAP65-1-induced microtubule-bundling activity	Salt stress signaling enhances tubulin polymerization involving *At*MAP65-1 increases salt tolerance by microtubule stabilization	*At*PLDα1	16:0–18:1PA, 16:0–18:2PA, 18:0–18:1PA, 18:0–18:2PA, but not di18:0PA	[74]
*At*MPK6 (At2g43790), a mitogen-activated protein kinase 6	-	Stimulation	Salt stress signaling, salt tolerance by *At*MPK6-dependent phosphorylation of *At*SOS1	*At*PLDα1	Significantly to di18:1PA, di18:2PA, 16:0–18:1PA, 16:0–18:2 PA, 18:0–18:2PA	[75]
*At*MKK7 (At1g18350), *At*MKK9 (At1g73500), a mitogen-activated protein kinase kinases	-	Activation, translocation to PM	Salt stress signaling, salt tolerance, maximization of the signal transduction efficiency	*At*PLDα1	Strongly to di18:1PA, 16:0–18:1PA, 16:0–18:2PA, 18:0–18:2PA	[76]
*At*MPK3 (At3g45640)/ *At*MPK6 (At2g43790), a mitogen-activated protein kinases	-	Stimulation, interaction with and phosphorylation of hypoxia-related transcription factor RAP2/12	Hypoxia stress signaling, translocation to nucleus and stimulation of hypoxia-related transcription factor RAP2/12, feedback inhibition of PA production, tolerance to submergence	*At*PLDα1, *At*PLDδ	-	[77]
*At*PINOID (At2g34650), a protein kinase	Lys119 to Lys121	Stimulation, tethering to PM	Salt stress signaling, salt tolerance, enhanced *At*PINOID-dependent *At*PIN2 phosphorylation and PM localization, activation of PIN2-mediated auxin efflux and redistribution	*At*PLDα1, *At*PLDδ	di18:1PA, di18:2PA, 16:0–18:1PA, 16:0–18:2PA, 18:0–18:1PA, 18:0-18:2PA	[65]
*At*SnRK2.4 (At1g10940)	-	Phosphorylation of SnRK2 targets, affects its interaction with PA and subcellular localization	-	-	16:0/18:1 PA	[78]
*Os*AKT2 (Os05g35410), a potassium channel	Arg190/Lys191, Arg644/Arg645, Arg755/Arg756 in ANK domain, cytoplasmic domain, S4 voltage sensor	Inhibition	Seedlings growth under short-day conditions	-	Strongly to 16:0–18:2PA, much weaker to di16:0PA, di18:0PA, di18:1PA, di18:2PA, 16:0–18:1PA, 18:0–18:1PA, 18:0–18:2 PA	[67,79]
*Os*GID1 (Os05g33730), a gibberellin receptor	Arg79, Arg82	Tethering to nucleus	Gibberellin signaling, promotion of *Os*SLR1 (a suppressor DELLA protein) degradation	*Os*PLDα6		[80]

Abbreviations: ABA, abscisic acid, Arg, arginine (R), Lys, lysine (K), NO, nitric oxide, PM, plasma membrane, PP2C, protein phosphatase 2C, ROS, reactive oxygen species.

**Table 2 ijms-23-03227-t002:** Co-expression analysis of *Arabidopsis thaliana* genes coding selected peptide receptors and enzymes of PA signaling performed in silico using “Genevestigator” database [55]. All statistical data were implemented and provided by “Genevestigator” database; 1031 samples, 25 anatomical parts, 691 perturbations (AT_mRNASeq_ARABI), 10615 samples, 3243 perturbations, 7 developmental stages (AT_AFFY_ATH1) were analyzed. Pearson’s correlation coefficient (score) was used for statistical analysis of gene co-expression. Database dataset denotes aggregated expression data according to a specific category (samples, anatomy, development, perturbation).

Bioactive Peptide	Receptor	Co-Expressed PA-Metabolizing Enzyme	Platform	Database Dataset	Statistical Notes (Score)
PSK	PSKR1	PLDζ1	AT_mRNASeq_ARABI	anatomy	0.9
PSK	PSKR1	PLDγ1	AT_AFFY_ATH1	perturbation	0.69
PSK	PSKR1	PLDγ1	AT_AFFY_ATH1	sample	0.69
CEP1	CEPR1	PI-PLC3	AT_mRNASeq_ARABI	anatomy	0.89
CLE25	BAM2	NPC6	AT_AFFY_ATH1	sample	0.68
PIP1/2	RLK7	DGK5	AT_mRNASeq_ARABI	perturbation	0.85
PIP1/2	RLK7	PLDγ1	AT_mRNASeq_ARABI	sample	0.79
PEP	PEPR1	DGK1	AT_AFFY_ATH1	sample	0.56
PEP	PEPR2	DGK5	AT_mRNASeq_ARABI	sample	0.64
EPF2	TMM	NPC2	AT_AFFY_ATH1	development	0.9
LURE1	MIDS1	PI-PLC6	AT_mRNASeq_ARABI	perturbation	0.61
LURE1	MIK2	PLDγ1	AT_mRNASeq_ARABI	perturbation	0.82
LURE1	PRK8	PLDα2	AT_AFFY_ATH1	sample	0.82
LURE1	PRK3	DGK4	AT_mRNASeq_ARABI	perturbation	0.74
LURE1	PRK5	DGK4	AT_mRNASeq_ARABI	perturbation	0.74
GRIM REAPER	PRK5	DGK4	AT_mRNASeq_ARABI	perturbation	0.74
ESF1	SSP	DGK4	AT_AFFY_ATH1	sample	0.75
RALF1	BAK1	PLDγ1	AT_AFFY_ATH1	perturbation	0.73
RALF1	BAK1	PLDγ1	AT_mRNASeq_ARABI	sample	0.8
RALF1	BAK1	PLDγ1	AT_mRNASeq_ARABI	anatomy	0.85
Ralf4/19	BUPS1	DGK6	AT_AFFY_AG	sample	0.9
Ralf4/19	BUPS1	PI-PLC6	AT_mRNASeq_ARABI	sample	0.92
Ralf4/19	BUPS2	PI-PLC6	AT_mRNASeq_ARABI	sample	0.95
Ralf1/17/23/ 32/33	FERONIA	NPC1	AT_mRNASeq_ARABI	sample	0.7
Ralf1/17/23/ 32/33	FERONIA	PLDα1	AT_mRNASeq_ARABI	anatomy	0.88
Ralf1/17/23/ 32/33	FERONIA	PLDα1	AT_mRNASeq_ARABI	sample	0.79
Ralf1/17/23/ 32/33	FERONIA	PLDα1	AT_AFFY_ATH1	sample	0.5
Ralf4/19/34	ANXUR2	PLDα2	AT_AFFY_AGRONOMICS	sample	0.83
CLE45	SKM1	PLDε	AT_mRNASeq_ARABI	anatomy	0.61
CLE45	SKM1	PI-PLC5	AT_mRNASeq_ARABI	anatomy	0.57
CLE42	PXL2	PLDδ	AT_mRNASeq_ARABI	perturbation	0.71
TPD1	EMS1	PLDε	AT_AFFY_AGRONOMICS	perturbation	0.82

Abbreviations: BAK1, BRI1-associated kinase 1; BAM, BARELY ANY MERISTEM; BUPS, BUDDHA’S PAPER SEAL; CEP, C-TERMINALLY ENCODED PEPTIDE; CEPR, C-TERMINALLY ENCODED PEPTIDE RECEPTOR; CLE, CLAVATA3/EMBRYO SURROUNDING REGION; DGK, diacylglycerol kinase; EMS1, EXCESS MICROSPOROCYTES1; EPF2, EPIDERMAL PATTERNING FACTOR 2; ESF1, EMBRYO SURROUNDING FACTOR1; MDIS, MALE DISCOVERER; MIK, MDIS1-INTERACTING RECEPTOR-LIKE KINASE; NPC, non-specific phospholipase C; PEP, PLANT ELICITOR PEPTIDE; PEPR, PLANT ELICITOR PEPTIDE RECEPTOR; PIP, PAMP-INDUCED SECRETED PEPTIDES; PI-PLC, phosphatidylinositol-specific phospholipase C; PLD, phospholipase D; PRK, POLLEN-SPECIFIC RECEPTOR-LIKE KINASE; PSK, PHYTOSULFOKINE; PSRKR, PHYTOSULFOKINE RECEPTOR; PXL, PHLOEM INTERCALATED WITH XYLEM-LIKE; RALF, RAPID ALKALINIZATION FACTOR; RLK, RECEPTOR-LIKE KINASE; SKM, STERILITYREGULATING KINASE MEMBER; SSP, SHORT SUSPENSOR; TMM, TOO MANY MOUTHS; TPD1, TAPETUM DETERMINANT1.

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
