# Peer review of "Phosphatidic Acid in Plant Hormonal Signaling: From Target Proteins to Membrane Conformations"

_ijms, 2022, doi:10.3390/ijms23063227_

Round 1
Reviewer 1 Report
Phosphatidic acids are an important part of cellular metabolism and signaling. In this review, Kolesnikov et al. have tried to address diverse aspects of PA role in plants. The manuscript requires some polishing, but I believe that it could be a valuable addition to the IJMS.
Major issues:
1) The authors should consider improving the structure of the manuscript. For instance, the fact that PA is not a discrete molecule and that there are differences in the biological activity of different PA species should be in the introduction (not on page 8). I believe that it could be also interesting to include some general comparison with well-established animal models of PA signaling and to list the concentration range of PAs in plants.
2) Figure 2
- The heat map representation would highly benefit from ANOVA and posthoc analysis to indicate significant differences in expression profiles.
- The authors present two different analyses based on outputs of RNA-seq and classical array experiments, respectively. These maps are different, yet this fact is not commented on or addressed in the manuscript, indicating that the differences are not relevant. If this is indeed the case, the authors should present only a single heat map.
- The authors should indicate how many experiments were used for this analysis.
- 'Percent of expression potential' - this term requires a definition.
3) Missing references.
There are missing references and the authors should reevaluate the available literature to include all relevant studies. For instance, the phythormonal part of the review is missing chapters for cytokinin and gibberellin (e.g., A rapid cytokinin response assay in Arabidopsis indicates a role for phospholipase D in cytokinin signalling; Phospholipase Dα6 and phosphatidic acid regulate gibberellin signaling in rice), information about lyso-PA acyl-transferases is missing (https://doi.org/10.1093/jxb/erab504), and the corresponding source manuscript should be cited for the bioinformatics analyses, not only GeneVestigator database (e.g., Figure 5).
4) Figure 3 - It is not clear how and why these five proteins were selected. I believe that it would be more illustrative to visualize proteins listed in Table 1.
Minor issues:
- LPP, PLD, DGK - abbreviation explained only in figure legends, not in the main manuscript
- Table 1 - incorrect decimal separator; missing information - how many experiments, employed statistics, legend (explain the meaning of Database dataset field)
- line 179-180 should be transferred into the next chapter
- Line 431 - hormonal regulation of PAs biosynthesis should not be listed in this chapter ('Phosphatidic acid as a signal transducer for other biologically active substances') this does not belong to that.
- Supplementary Figure 1. This figure is illegible and mislabeled (Figure 6) or misplaced.
- H202 is with O, not zero
Author Response
We are sincerely grateful to the reviewers for their suggestions and comments, all of them are taken into account, moreover, while the article was in our review, an article on autophagy and PLD went out of print, we -used in the conclusion

Reviewer 2 Report
The manuscript entitled "Phosphatidiс acid in plant hormonal signaling: from target proteins to membrane conformations" by Kolesnikov et al., was reviewed for publication in IJMS (manuscript 1535972). The manuscript is excellent, gives a thorough review of the area related to signalling through phosphatidic acid, and presents a comprehensive overview and new insight into the subject in relation to plant hormone signalling mechanisms. My main criticism is with some of the Figures, the quality of which could be improved. I made a few corrections and comments on the pdf to facilitate revisions, and a few comments below.
Figure 1, could the quality of this figure be improved? The text appears squished, and the quality of the chemical structures is blurry.
Figure 2, could the quality of this figure be improved? The fonts are very small for all labels, and the images for the different developmental stages are very small, all could be improved by making larger.
Lines 216-222, this paragraph seems to be a bit out of place, since it comes after a thorough discussion about PA-binding proteins and binding sites. Maybe this could be put at the top of the section to clarify that at the outset that there are no common motifs known in plant or animal proteins, but…….. Or, start this paragraph off with a better lead in than “Since….”.
Figure 4, there are errors to correct I indicated directly on the pdf.
Figure 5, font very small, could be improved for better readability.

Author Response

(The authors gave the same response as above.)

Round 2
Reviewer 1 Report
The authors did not provide a point-by-point response to the review, and the reason for skipping some of my comments is thus not clear. For instance, references to cytokinin and PA are missing, lysophosphatidic acid acyltransferase is not included, Figure 2 contains two different heatmaps, and original data resources are not cited in the GeneVestigator-based data mining.
However, the revised version of the manuscript addressed some of my concerns, and I don't have any other input that would improve this work.
Author Response
Dear Sirs! We are very grateful to you for helping with our manuscript. Your advices and suggestions significantly improved the text readability and figures quality.
- We placed the information on the biological activity of different PA species in introduction of the article, added some general comparisons of plant PA signaling with well-established animal models of PA signaling, and also added information on PA concentrations in plants also to the introduction. Information of the concentration of PA in plants was added to the “Introduction”.
- “Suppl. Figure 1” was re-labelled to “Fig. 6” and “Suppl. Table 1” – to “Table 2”. The heat map representation in Fig. 5-6 included statistical analysis when the genes were analyzed in “Genevestigator” database, and the results obtained were chosen and taken to the review. In Fig. 6 we leaved the heatmap with classical array analysis, because there were the most number of known biologically active substances analyzed. Therefore, in Fig. 2 we also leaved the heatmap with classical array analysis. We also described how many experiments were used for the analysis for the Figure 2 and 6, and defined the term “percent of expression potential” in the Legend to the Figure 2.
- We aralyzed all relevant studies and included all missing references devoted to the role of phosphatidic acid in gibberellin signaling, cytokinin and lysophosphatidic acid acyl transferase.
- Proteins in Figure 3 were selected as they are PA targets that participate in signal transduction (among all of the known plant PA-binding proteins). It was shortly explained in the text (Lines 220-222). However, in order to make this explanation wider and understandable, we added restructured and supplemented the first sentence in the legend of the Figure 3.
Minor issues. We added abbreviations PLD, DGK, and LPP to the text, added the number of experiments to the Table 1, statistics, legend, and explain the meaning “Database dataset”, change decimal separator (“,” was changed to “.”). As a result, the Table 1 started to be more understandable. We also transferred lines 179-180 to the next Chapter, and also transferred Line 431 to the “Hormonal” chapter to the appropriate place. Fig. 6 was also improved. The “H202” was corrected to “H2O2” in Line 421. All these changes in text were marked in blue.
We also made some corrections in text (marked in red).
Please see the attachment
